# The Haemodynamic and Pathophysiological Mechanisms of Calcific Aortic Valve Disease

**DOI:** 10.3390/biomedicines10061317

**Published:** 2022-06-03

**Authors:** Lydia Hanna, Chlöe Armour, Xiao Yun Xu, Richard Gibbs

**Affiliations:** 1Imperial Vascular Unit, St Mary’s Hospital, Imperial College Healthcare NHS Trust, London W2 1NY, UK; 2Department of Surgery and Cancer, Imperial College London, London SW7 2AZ, UK; 3Department of Chemical Engineering, Imperial College London, London SW7 2AZ, UK; chloe.armour13@imperial.ac.uk (C.A.); yun.xu@imperial.ac.uk (X.Y.X.)

**Keywords:** calcific aortic valve disease, aortic stenosis, valvular endothelial cells, valvular interstitial cells

## 1. Introduction

The aortic valve (AoV) is the outflow valve for the left heart. It opens and closes > 100,000 times a day and maintains unidirectional antegrade blood flow from the left ventricle into the aorta without regurgitation of blood, at a peak value of 1.35 ± 0.35 m/s [1]. Calcific aortic valve disease (CAVD) is a chronic, slowly progressive disorder ranging from aortic sclerosis, a mild thickening of the AoV due to accumulation of calcium deposits on the AoV without obstruction to blood flow, through to aortic stenosis, severely calcified and thickened valve leaflets that become stiff and narrowed (Figure 1). At this stage, leaflet motion is impaired and restricted, leading to left ventricular outflow obstruction and reduced blood flow through the narrowed AoV [2]. Significantly increased mechanical stress is placed on the left ventricle as the pressure gradient between the left ventricle and the aorta increases. As a result, the left ventricle hypertrophies, cardiac output decreases, and there is an increased risk of life-threatening complications such as heart failure, aortic dilatation and aneurysm formation, aortic dissection, and sudden cardiac death [2,3]. On a micro-cellular level, the AoV is subject to endothelial damage, inflammation, disruption and remodelling of the three-layered extracellular matrix (ECM) that comprises the AoV [4].

CAVD is the third commonest heart disease in the western world. The prevalence of CAVD increases significantly with advancing age and is about 10% over the age of 80 [5,6]. Furthermore, with increasing life expectancy in the developed world, CAVD now constitutes a growing global health burden with a reported 3–4-fold increase in incidence and prevalence over the last 30 years [7].

The association with increasing age has led many to classify CAVD as a passive ‘degenerative’ process due to wear-and-tear of the leaflets. Other risk factors for the development of CAVD have included those commonly associated with atherosclerosis, including male gender, cigarette smoking, hypercholesterolemia particularly of low-density lipoproteins (LDLs), hypertension and diabetes mellitus [8].

However, there is increasing evidence that CAVD is an active disease entity characterized by a cascade of cellular and molecular changes in response to haemodynamic and shear stress abnormalities, endothelial damage, inflammation, lipoprotein deposition, and genetic factors that ultimately result in calcification and ossification of the AoV [4,9,10]. It is this calcification and ossification that distinguishes CAVD from atherosclerosis that by comparison is defined by the presence of vulnerable plaques, despite early lesions of CAVD resembling atherosclerosis.

Under specific stimulators, valvular endothelial cells (VECs) and valvular interstitial cells (VICs), two cell types that are native to the AoV and integral to AoV homeostasis, are thought to develop a pro-calcific phenotype and contribute to CAVD. The exact pathways that culminate in aortic sclerosis and stenosis remain incompletely understood. To date, there is no medical therapy to prevent or halt the calcification process, and the only current treatment option is surgical replacement of the AoV or transcatheter aortic valve replacement (TAVR). A greater understanding of the mechanisms of CAVD may yield disease-specific therapeutic targets. This review will focus on the haemodynamic stimuli and downstream cellular and molecular mechanisms that contribute to the development of CAVD.

## 2. Anatomy and Histology of the Normal Aortic Valve

The AoV separates the left ventricle of the heart from the aorta. The normal healthy AoV is made up of three leaflets. Two of these leaflets have outflows to the coronary arteries and are known as the left and right coronary leaflets. The third leaflet is known as the non-coronary leaflet. Each leaflet is arranged in three layers and comprises an extracellular matrix (ECM) of varying degrees of elastin, collagen and glycosaminoglycans and two heterogenous cell types (VECs and VICs). The layer facing the ventricle is known as the ventricularis, and that facing the aorta is known as the fibrosa. Sandwiched in between the two is the spongiosa (Figure 2).

### 2.1. Layers of the Aortic Valve

The ventricularis layer is made up of elastin fibres aligned in a radial direction that facilitates changes in the shape of the leaflets to allow fast and consistent compression of the leaflets during opening and closing, ensuring correct valve closure [11]. The fibrosa is made up of circumferentially arranged type I and III collagen fibres that provide tensile strength to the leaflets [4,12]. The spongiosa is made up of a matrix of proteoglycans (PGs) and glycosaminoglycans (GAGs) that lubricate and reduce the shear between the ventricularis and fibrosa layers during valve deformation and act as ‘shock absorbers’ to resist the mechanical stress generated during the cardiac cycle [13]. All three layers are avascular, devoid of cellular infiltrates such as lymphocytes and monocytes, and are innervated by adrenergic and cholinergic nerve fibres, extracting nutrients from the surrounding blood [14,15].

### 2.2. Valvular Endothelial Cells (VECs)

VECs line the ventricularis and fibrosa layers only, in continuity with the ventricular endocardium and the aortic endothelium, and are in direct contact with blood and the shear forces associated with its flow through the AoV (Figure 3). Unlike vascular endothelial cells, VECs are aligned perpendicularly to the direction of blood flow. Laminar flow has been found to induce mechanotransduction pathways unique to VECs that lead to cytoskeletal reorientation [16,17,18]. As such, VECs are responsible for the translation of a variety of mechanical stimuli (alterations in flow patterns and subsequent changes in shear stress, pressure, stress and bending of the AoV) into biological responses (mechanotransduction) through a variety of cellular and molecular mechanosensors including transmembrane proteins, ion channels and cytoskeleton proteins, all of which lead to structural changes in the AoV leaflet and allow it to function and adapt in a harsh haemodynamic environment [19]. VECs are also responsible for the regulation of adhesion of inflammatory and proliferative cells, paracrine signalling [20], and VEC–VIC interaction [21,22].

### 2.3. Valvular Interstitial Cells (VICs)

VICs are found in each layer within the ECM. VICs comprise fibroblasts and vascular smooth muscle cells and are responsible for the synthesis of the ECM during valvulogenesis [15]. During adult life, VICs are in a quiescent state akin to quiescent fibroblasts (qVICs) and balance the production and degradation of the ECM [23]. Upon an adequate stimulus, qVICs differentiate into activated myofibroblasts (aVICs) to regulate remodelling of the ECM, dedifferentiating back to qVICs on cessation of this stimulus [24,25].

Under certain stimuli, most notably aberrant blood flow patterns and shear stress, VECs and VICs have the capacity to undergo transformation from physiological cell types integral to the normal functioning of the AoV and into a pathological phenotype through a variety of cellular and molecular responses that culminate in AoV calcification as summarized in Figure 4 and will be discussed herein.

## 3. The Impact of Aortic Valve Haemodynamics on Molecular Pathways That Lead to Calcification of the Aortic Valve

Calcification of the AoV follows a specific pattern with respect to the layer and leaflet affected. The early focal subendothelial lesions of CAVD and leaflet calcification are side-specific with a predilection for the fibrosa layer [4]. Similarly, the non-coronary leaflet (leaflet without coronary outflow) is the leaflet most likely to calcify [27]. Flow at the fibrosa side and non-coronary leaflet is oscillatory and of low-shear compared to the ventricular side and at the coronary leaflets where high-shear laminar flow predominates [28,29].

Finite-element analysis with in vitro cell culture studies has shown that endothelial cells exposed to shear stress waveforms within the ventricularis demonstrated increased expression of ‘atheroprotective’ transcription factors (e.g., Kruppel-like factor 2, KLF2) compared to endothelial cells exposed to shear stress waveforms in the fibrosa [30]. Conversely, low levels of KLF2 have also been found in low and oscillatory shear stress conditions (±5 dynes/cm^2^) of the fibrosa, whereas higher levels of KLF2 were found in high unidirectional shear stress of the ventricularis (20 dynes/cm^2^) [31].

Tissue transforming growth factor (TGF)-β1 has been found to be higher in patients with aortic stenosis compared to healthy controls [32,33]. The addition of TGF-β1 to sheep aortic valve interstitial cell cultures has been found to induce the formation of apoptotic alkaline phosphatase (ALP) enriched nodules as well as their calcification [34]. Exposure of the fibrosa to non-physiological pulsatile shear stress (altered haemodynamics) was found to lead to increased expression of TGF-β1 in addition to bone morphogenic protein (BMP-4), an initiator of calcification, as well as the inflammatory markers vascular adhesion molecules vascular cell adhesion molecule 1 (VCAM-1) and intercellular adhesion molecule 1 (ICAM-1), which are known to promote the migration and adhesion of inflammatory cells to endothelial cells [35]. While oscillatory shear stress, normally encountered on the fibrosa side, did not induce these responses, neither pulsatile shear stress nor oscillatory shear stress induced these responses on the ventricularis side [35]. Interestingly, BMP-4 was found to be downregulated when porcine aortic valve cells were exposed to laminar shear stress [36]. Cyclical strain has also been found to cause valve calcification. Balachandran et al. demonstrate a magnitude-dependent increased expression of BMP2 and 4 as well as suppression of runt-related transcription factor 2 (RUNX2), an osteoblast stimulator, on the fibrosa in response to elevated stretch [37].

The NOTCH signalling genes play an important role in the development of the cardiovascular system during embryonic life. Laminar shear stress has been found to activate NOTCH1 in human aortic VECs, which led to downregulation of osteoblast-like genes [38]. Mutations in NOTCH1 have resulted in severe AoV calcification due to impaired suppression of RUNX2 [39]. Furthermore, porcine aortic VICs treated with NOTCH1 inhibitors showed increased calcification compared to wild-type cells [40].

Endothelial-to-mesenchymal transition (EndMT) is the process whereby VECs transform into mesenchymal cells. The EndMT process underpins valvulogenesis in the embryo; however, during adult life, EndMT may contribute to CAVD. Using sheep VECs, Balachandran et al. have demonstrated that reprogramming of VECs into mesenchymal cells occurs in response to cyclical mechanical strain [41]. The authors also observed a differential downstream signalling response whereby low-strain EndMT occurred via increased TGFβ1, whereas high-strain EndMT occurred via wnt/β signalling, a pathway that has been linked to aortic valve calcification [42]. Mahler et al. also found that VECs exposed to oscillatory shear stresses underwent EndMT (increased expression of EndMT markers ACTA 2, Snail, TGF-β1, α-SMA and inflammatory proteins ICAM1 and NFKβ1) known to be expressed in the endothelium of CAVD, when compared to control VECs exposed to high steady shear [43]. Similarly, isolated porcine aortic valve endothelial cells exposed to cyclic strain resulted in upregulation of VCAM1, ICAM1 and E-selection cells [44].

Endothelial nitric oxide synthase (eNOS) is a shear-sensitive gene and is a potent vasodilator. eNOS is upregulated in response to laminar shear as part of the physiological response to maintain homeostasis of the AoV and blood vessels and is downregulated in low and oscillatory shear conditions seen in pathological disease states [45]. Among the 30 main metabolites identified in patients with aortic stenosis, 17 were found to be related to NO metabolism [46]. It has been suggested that inhibition of fibrosa calcification may be the result of diffusion of NO from the ventricularis surface and that this process is hindered at the early stages of CAVD prior to calcification [4]. Using porcine aortic valves, Richard et al. demonstrated that the ventricularis had elevated levels of cGMP expression, a surrogate for NO, compared to the fibrosa. In the same study, pulsatile, unidirectional shear stress typical of the ventricularis led to significantly more cGMP production than oscillatory shear stress patterns typical of the fibrosa [47]. Furthermore, higher osteocalcin was observed on the fibrosa side when the receptor for NO (sGC) was blocked.

MicroRNAs (miRNAs) are regulators of gene and protein expression and have also been implicated in CAVD in response to mechanical stimuli. Through microarray and bioinformatics analysis, Rathan et al. [48] were able to isolate miR-214 using porcine aortic valve leaflets in an ex vivo shear system. Under oscillatory shear stress conditions, higher expression of miR-214 and increased thickness and calcification of the fibrosa were observed in comparison to the ventricularis via a miR-214–TGFβ1 pathway. Interestingly, inhibition of miR-214 did not affect OS-induced calcification, with the authors postulating that this may be because miR-214 and its target genes may be influenced by other mechanical stimuli such as stretch, bending and cyclic strain.

Heath et al. [49] investigated miRNA-181b. Human aortic valve endothelial cells from the fibrosa and the ventricularis from heart transplant recipients were exposed to unidirectional steady sheer stress (LS) and bidirectional oscillatory shear stress. miRNA-181b was found to be significantly higher at the fibrosa under oscillatory shear stress and led to a downregulation of TIMP3, an inhibitor of matrix metalloproteinase activity (MMP), high levels of which are through to lead to ECM degradation and hasten sclerosis and calcification in CAVD.

Esmerats et al. found upregulation of ubiquitin E2 ligase C (UBE2C) in human aortic VECs through miR-483 exposed to oscillatory flow. UBE2C was also responsible for endothelial inflammation and EndMT by increasing the hypoxia-inducible factor-1α (HIF1α) level by ubiquitinating and degrading its upstream regulator, von Hippel-Lindau protein (pVHL). The same study group isolated an miR-483 mimic that was found to reduce EndMT and inflammation in human aortic valve endothelial cells and calcification of porcine AV leaflets through downregulation of UBE2C. Furthermore, treatment with PX478, an HIF-1α inhibitor, significantly reduced porcine aortic valve calcification in static and oscillatory flow conditions [50]. In recent years, there has been a focus on miRNA inhibition. Toshima et al. [51] recently demonstrated that inhibition of microRNA-34a ameliorated calcification in porcine aortic VICs through modulation of NOTCH1-RUNX2 signalling. Similarly, Yang et al. [52] demonstrate that miRNA-34c inhibits osteogenic differentiation of human VICS obtained from CAVD patients through suppression of STC1 and JNK signalling pathways.

## 4. Downstream Molecular and Cellular Pathways Leading to Calcific Aortic Valve Disease

### 4.1. Lipidogenesis

The earliest lesion of CAVD appears as a plaque composed of extracellular lipids in the subendothelial region, akin to atherosclerosis. The elastic lamina is displaced, and the plaque extends into the fibrosa [4]. Lipoproteins are particularly common, leading some to conclude they originate from the plasma [53]. The valvular endothelial dysfunction (e.g., initiated by mechanical stimuli) observed in CAVD is thought to result in the loss of the barrier function of endothelium [9,10]. This may then permit infiltration of lipids to the subendothelial region. Hypercholesterolemia and an excess of lipoprotein a (Lp(a)) are both well-described risk factors for CAVD, giving further credibility to this theory [54].

On histological analysis, oxidised low-density lipoproteins (Ox-LDL) increased along with T cells and macrophages are found in abundance close to calcium nodules in the subendothelial layer of the fibrosa [53]. High levels of plasma lipoproteins are thought to lead to CAVD through oxidation. Lipoproteins play a crucial role in carrying oxidised phospholipid (oxPL), which can activate endothelial cells, leading to an inflammatory infiltrate [55,56] that in turn may trigger the production and release of a variety of factors that stimulate VICs to acquire an osteogenic profile [4,53], as will be described later. These factors include TLR2 [57], LPC [55] and ATX [58], all of which have been found in abundance in patients with CAVD. Despite the convincing role of lipids in the pathogenesis of CAVD, randomised controlled trials (SALTIRE, SEAS, TASS, ASTRONOMER) have all failed to show that statins can prevent or regress CAVD [59,60,61,62]. While statins inhibit HMC-CoA reductase and may reduce plasma LDL levels, the negative findings in these trials may be because statins increase the levels of Lp(a) [63].

### 4.2. Vitamin K

Vitamin K is a lipid-soluble vitamin that functions as a cofactor for the activation of a wide variety of vitamin K-dependent proteins. Matrix G1a protein (MGP) is a vitamin K-dependent protein that also plays an important role in the inhibition of vascular calcification by inhibiting BMP-2 [64]. Significantly lower levels of MGP have been found in human AoV VICs compared to normal VICs [65]. Accordingly, in their single-centre open label randomized trial comparing 2 mg phytomenadione to placebo, Brandenburg et al. [66] observed greater progression in the aortic valve calcification score in the placebo group (*n* = 34) compared to the intervention arm (*n* = 38). Larger trials are, however, required to confirm these findings prior to the widespread use of vitamin K in clinical practice.

### 4.3. Inflammation

In addition to LDLs, early lesions of CAVD are also composed of inflammatory cells, including macrophages and foam cells [4], T and B cells [4,67], and mast cells [68]. Recruitment of inflammatory cells into the early lesion is thought to be due to Ox-LDL initiated upregulation of cell adhesion molecules such as ICAM1, VCAM1 and E-selection that would otherwise not be expressed on VECs under normal conditions, in addition to damaged endothelial cells. These processes ultimately permit the recruitment and trans-endothelial migration of several inflammatory cells [69,70]. It is widely recognized that inflammatory cells release several proinflammatory and pro-osteogenic molecules that contribute to the osteogenic profile differentiation of VICs and that are responsible for ECM degradation/remodelling, all of which has been implicated in the pathogenesis of CAVD. These include TNFa, IL1B, IL-6, RANKL, MMPs, oligoclonal CD4+ and CD8+ T cells. Receptor Activator of Nuclear factor Kappa B Ligand (RANKL) and osteoprotegrin (OPG) are two noteworthy inflammatory molecules. Both are known to be involved in bone turnover and vascular calcification. In addition, they have been found in abundance in calcific human aortic valves with the ability to stimulate the differentiation of VICs into an osteoblastic phenotype [71]. Denosumab, an IgG_2_ monoclonal antibody and RANKL inhibitor, was postulated to be a potential medical treatment that may halt CAVD. However, the SALTIRE2 trial found that Denosumab had no effect of disease progression of aortic stenosis as assessed by echocardiography, computed tomography or ^18^F-sodium fluoride positron emission tomography [72]. Alendronic acid, a bisphosphonate used to decrease bone osteoclastic activity in osteoporosis, similarly had no effect [72].

Macrophages and VICs have also been found to release extracellular vesicles (EV), membrane vesicles secreted from cells that contain intracellular contents, and have been implicated in microcalcification in AoV disease [73]. Under pathological conditions, calcifying EVs have been found to form spherical microcalcifications that may fuse together to form macrocalcifications and cause dysfunction of the valves [74].

### 4.4. Angiotensin Converting Enzyme (ACE)

Angiotensin converting enzyme (ACE) has also been implicated in the pathogenesis of CAVD. High levels of ACE and angiotensin II, as well as AT1R, a receptor for angiotensin II, not otherwise found in normal AoVs, have all been found in human stenotic AoVs [75]. Angiotensin II is thought to promote inflammation and fibrosis in AoVs through many downstream signalling pathways [76]. Interestingly, while some ACE may be produced locally, or colocalizes with macrophages, the majority (including angiotensin II) is ‘carried’ into the subendothelial lesion via LDL (apolipoprotein B) protein in cholesterol particles [75]. Binding of angiotensin II to the AT1R leads to expression of several inflammatory markers such is ICAM1 and VCAM1 [77]. Inhibition of ATR1 with Olmesartan was also found to inhibit osteoblast differentiation in the AoVs of hypercholesterolemic rabbits [78]. Similarly, the use of ACE inhibitor (ACEI) in patients has been found to reduce AoV calcium accumulation under electron beam computed tomographic scans [79] and reduced the progression of aortic sclerosis [80]. However, other studies have found no difference in haemodynamic progression of aortic stenosis with the use of ACEI [81].

### 4.5. Reactive Oxygen Species

Reactive oxygen species (ROS) are chemical forms of molecular oxygen that exist as free radial species with at least one free electron such as superoxide, and nonradical species with two electrons such as hydrogen peroxide. Their physiological functions include the regulation of cell growth and differentiation and the modulation of several metabolic processes; however, their pathological role can lead to DNA damage and dysfunctional DNA repair mechanisms [82,83]. A key feature of ROS is their ability to oxidise other molecules such as LDLs [84]. Interestingly, in bovine VECs, an increase in oxidised LDL has been found to correlate with the upregulation of NADPH oxidase, an important complex involved in the generation of superoxide [85]. High levels of ROS including superoxide and hydrogen peroxide have been found in calcified human aortic valves [86,87,88,89]. Exogenous ROS have been found to promote calcification of VICs through RUNX2 and other pro-osteogenic signalling pathways [87] as well as VCAM1 expression [89]. Furthermore, ROS-inhibiting treatments have been found to decrease calcium nodule formation in porcine and rabbit VICs [89,90,91], reduce ROS-induced DNA damage in VICs identified in stenosed AoV from CAVD patients [87] and dampen the TNF-alpha mediated endothelial inflammatory response, myofibroblast activation, calcification and ECM degradation [89].

## 5. Valvular Endothelial Cell to Valvular Interstitial Cell Interactions in the Pathogenesis of Calcific Aortic Valve Disease

VECs and VICs are the key effector cells in AoV physiology, but also in pathophysiological conditions such as CAVD, when both cell types may exhibit pro-calcific behaviour. To date, there is no definitive direct link between VECs and CAVD. However, several studies would seem to suggest the pro-osteogenic role of VECs in CAVD. VECs on the fibrosa express fewer inhibitors of calcification compared to the ventricularis side [92]. As highlighted above, abnormalities in physiological shear stress are likely to account for this phenotypic heterogeneity that can then activate a cascade of cellular, molecular, and inflammatory pathways, including EndMT, that ultimately result in aortic calve calcification. Hjortnaes et al. also demonstrated the osteogenic differentiation of VECs via EndMT in stenotic human aortic valves with significant increases in ALP, TGFB1, and RUNX2. The same group were also able to propose that VEC may act as a reservoir for VICs with the propensity to develop into osteogenic VICs [93].

VICs are the predominant cell type in the AoV. In addition to qVICs, VICs can also differentiate into activated myofibroblast-like VICs (aVICs) expressing alpha smooth muscle actin (alpha-SMA) responsible for ECM remodelling, and an osteogenic phenotype (oVIC), which is the prevailing cell that deposits calcium in the AoV. The mechanisms that lead to the differentiation of oVIC remain poorly understood. However, as mentioned in the preceding sections, inflammatory markers such as TGF-B, IL-6, TNF, and NFKB have all been implicated [94]. In turn, calcification induced by VICs is thought to involve BMP2-mediated ALP expression [95].

The interactions between VECs and VICs and their impact on valve integrity remain of great interest. The study by Hjortnaes et al. [93] demonstrated that VICs could prevent the osteogenic differentiation of VECs, but VECs could not prevent the formation of oVICs. Conversely, Richards et al. [47] demonstrated that porcine VECs inhibited VIC-mediated calcification by inhibiting myofibroblastic activation and osteogenic differentiation. These findings serve to highlight the complexity of the inter-cellular pathways that remain incompletely characterized.

## 6. The Role of the Extracellular Matrix

In addition to lipid deposition (early lesion) and calcification (late lesion), thickening and fibrosis of the AoV is another histological feature of CAVD. Prior to the development of thickened and fibrotic valves, it is likely the ECM undergoes degradation followed by (abnormal) remodelling. However, as previously discussed, high levels of MMPs have been found in CAVD [96,97,98]. VICs, macrophages and monocytes are thought to be one source of these ECM-degrading enzymes [98,99]. Paradoxically, increased expression of tissue inhibitor of metalloproteinases (TIMPS) has also been found in CAVD [97,99]. At some point in the disease process, probably under the influence of inflammatory markers and other cytokines, MMP expression surpasses that of TIMP, shifting the balance towards ECM degradation [97]. Proteases including cathepsins have also been detected in high levels close to the sites of valvular angiogenesis, suggesting that ECM degradation by these enzymes facilitates the growth of new blood vessels [100].

Thickening and fibrosis of the AoV are due to unregulated and disorganised deposition of ECM components such as collagen, resulting in overgrowth and scarring of the valve [101]. TGF-B has been implicated in this process with a sustained production of wide and dense collagen fibres in the spongiosa layer compared to shorter fibres found in the fibrosa layer [102]. These changes are likely to be mediated by signalling pathways related to VICs.

## 7. Future Perspectives

### 7.1. Clinical Translation of Aortic Haemodynamics for the Diagnosis and Prognostication of Calcific Aortic Valve Disease

Recent developments in 3-dimensional (3D) phase-contrast magnetic resonance imaging (PC-MRI) with 3-directional velocity encoding, known as 4D flow MRI, now afford us the ability to directly quantify advanced haemodynamic parameters such as wall shear stress and pressure gradients in a wide range of cardiovascular diseases, including AoV disease, that may in the future aid disease diagnosis and prognostication and guide the timing of surgical intervention, beyond conventional imaging techniques. Archer et al. [103] note that 4D flow-derived pressure gradient across the AoV was more consistent with invasive pressure gradient measures than Doppler echocardiography. The pressure gradient change in 4D flow was also found to correlate better with left ventricular mass regression, an independent predictor of improved long-term survival [104]. Furthermore, both pressure gradient and effective orifice assessment of the valve by 4D flow were better associated to exercise tolerance as assessed by the New York Heart Association classification (NYHA) of heart failure and the six-minute walk test (6MWT), both objective measures that are increasingly used to guide the timing of AoV intervention [103]. Binter et al. [105] demonstrate that 4D flow turbulent kinetic energy (TKE), a measure of energy loss in disturbed flow such as that seen in AS not otherwise measured by echocardiography, provided valuable information regarding aortic stenosis severity. The abnormal flow patterns generated by aortic stenosis and the resultant stress on the left ventricle can also be visualized and quantified on 4D flow. Von Knobelsdorff-Brenkenhoff et al. [106] observed greater helical and vortical flow formation and eccentricity with elevated peak systolic shear stress in patients with aortic stenosis compared to healthy controls. Stronger vortical flow formations were associated with a high left ventricular mass index. In perhaps the largest study (*n* = 571) to use 4D flow in aortic stenosis patients, Ooji et al. [107] were able to deduce wall shear stress (WSS), the measure that is consistently demonstrated to influence endothelial cell function and valvular remodelling, and demonstrated that moderate to severe aortic stenosis increased WSS magnitude and variability in the ascending aorta. While these studies suggest a potential link between the resultant abnormal haemodynamics following aortic stenosis and impaired left ventricular function, there remains a paucity of longitudinal studies that have demonstrated a link between haemodynamic measures and progression of CAVD and whether the use of haemodynamic measures has the potential to influence clinical management for patients. These are certainly areas for future research.

### 7.2. Pharmacological Clinical Trials for Calcific Aortic Disease

While several pharmacological clinical trials have yielded disappointing results, the search for an effective medical therapy to combat CAVD continues. Table 1 summarises the ongoing clinical trials in CAVD.

## 8. Conclusions

Calcific aortic valve disease (CAVD) is a multifactorial disease. The complex flow patterns that surround the valve are the likely stimulus that sets off a cascade of cellular, molecular, and inflammatory pathways that may be exacerbated by risk factors, and that ultimately lead to calcification of the valve. These cell-signalling pathways are potential therapeutic targets that may in the future prevent the development and progression of the disease and abolish the need for surgical replacement of the diseased valve.

## Figures and Tables

**Figure 1 biomedicines-10-01317-f001:**
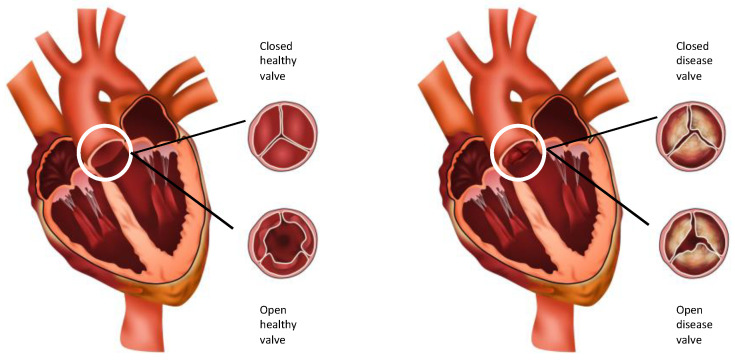
Diagrammatic representation of healthy aortic valve in closed and open state and a diseased aortic valve in closed and open state.

**Figure 2 biomedicines-10-01317-f002:**
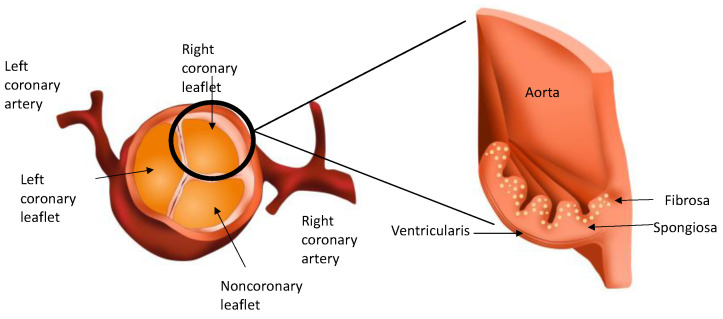
Diagrammatic representation of the coronary leaflets and a leaflet in cross-section showing the three layers of the aortic valve.

**Figure 3 biomedicines-10-01317-f003:**
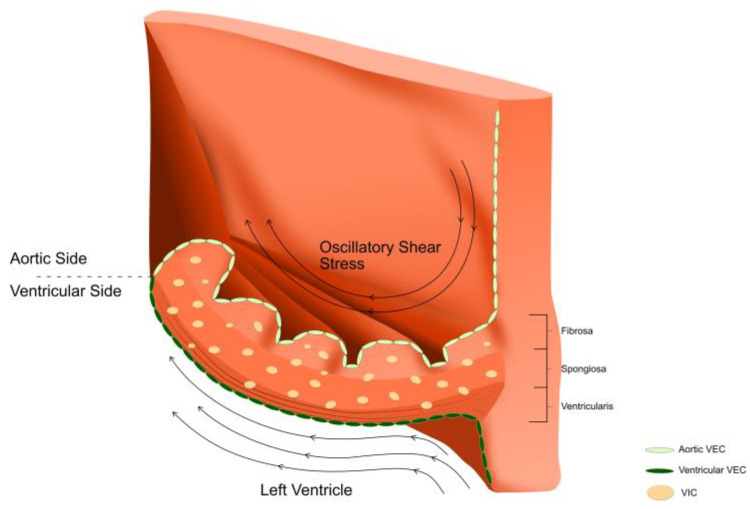
An aortic valve leaflet comprises an extracellular matrix that is organised into three layers; the ventricularis is on the ventricular side, the fibrosa is on the aortic side and the spongiosa is sandwiched in between these two layers. Valvular endothelial cells (VECs) line the ventricularis and fibrosa. Valvular interstitial cells (VICs) are found in all the layers. The aortic valve is exposed to a complex and harsh environment; the ventricularis side is exposed to laminar flow and shear stress, whereas the fibrosa side is exposed to oscillatory flow patterns and lower shear stresses.

**Figure 4 biomedicines-10-01317-f004:**
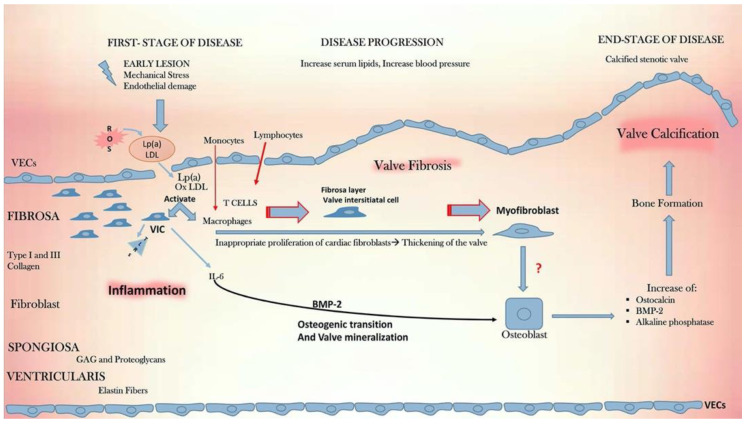
Cellular, molecular, and inflammatory cascades in CAVD. Figure reproduced with permission from Alushi et al. [26].

**Table 1 biomedicines-10-01317-t001:** Ongoing clinical trials targeting molecular pathways involving patients with calcific aortic valve disease (CAVD).

Study Name and NCT Identifier	Principal Investigator and Institution	Pharmacological Target and Treatment Arms	Recruitment Status
A study evaluating the effects of Ataciguat (HMR1766) on aortic valve calcification (CAVS) NCT02481258	Jordan D Miller; Mayo Clinic, Rochester, Minnesota, United States	Nitric oxide-independent soluble guanylate cyclase activator Ataciguat (HMR1766) 200 mg daily for 12 months vs. placebo	Completed, results awaited
Effect of PCSK9 inhibitors on calcific aortic valve disease (EPISODE) NCT04968509	Zhijian Wang; Beijing Anzhen Hospital, China	PCSK9 inhibitors (lipid lowering) 140 mg PCSK9 inhibitors (140 mg Evolocumab or Alirocumab subcutaneously every two weeks and conventional lipid lowering therapy (Atorvastatin 40–80 mg or Rosuvastatin 20–40 mg or Atorvastatin 20 mg+ Ezetimibe 10 mg or Rosuvastatin 10 mg+ Ezetimibe 10 mg vs. Conventional lipid-lowering therapy (Atorvastatin 40–80 mg or Rosuvastatin 20–40 mg or Atorvastatin 20 mg+ Ezetimibe 10 mg or Rosuvastatin 10 mg+ Ezetimibe)	Recruiting
A Study to evaluate the efficacy and safety of DA-1229 (Evogliptin) in patients with calcific aortic valve disease with mild to moderate stenosis (EVOID-AS) NCT05143177	REDNVIA Co., Ltd.	DA-1229 anti-diabetic Three arm study where patients will receive one of the three treatments orally once daily for period of 104 weeks; DA-1229 5 mg once vs. GroupDA-1229 10 mg vs. Placebo	Not yet recruiting
Colchicine and Inflammation in Aortic Stenosis (CHIANTI) NCT05162742	Radboud University Medical Centre	Colchicine vs. placebo	Not yet recruiting
Slower progress of calcification with Vitamin K2 (SLOW) NCT04429035	First Department of Cardiology, University of Athens Athens, Attiki, Greece Hippokration Hospital 1st Department of Cardiology, National and Kapodistrian University of Athens, Medical School Athens, Attiki, Greece	Dietary Supplement: Vitamin K2 vs. Dietary Supplement: Placebo	Recruiting

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
