# Peer review of "The Haemodynamic and Pathophysiological Mechanisms of Calcific Aortic Valve Disease"

_biomedicines, 2022, doi:10.3390/biomedicines10061317_

Round 1

Reviewer 1 Report

The authors present a review of pathological processes leading to aortic stenosis.

The review is comprehensive and details recent reports studying the condition.

I would encourage the authors to provide more figures describing the pathophysiological process.

The manuscript reviews the pathophysiology of calcific aortic stenosis, with potential pathways that can be addressed for prevention of aortic stenosis. Since aortic stenosis is the most common valvular disease, prevention of the condition is important albeit currently there is no proven preventive therapy. The review is important to identify potential targets for prevention of this condition. I would encourage the authors to provide more figures that depict the various pathophysiological processes

Author Response

We have now added in more pictures/diagrams that we have drawn. We hope they will be satisfactory to the reviewer. 

Reviewer 2 Report

The paper of Lydia Hanna et al is a review, discussing The Pathophysiological Mechanisms of Calcific Aortic Valve Disease and Aortic Stenosis. The review is divided into 8 sections including the introduction and the conclusion. Some of the sections are too long, I would suggest further sectioning.  The valve consist of two cell types valve interstitial cells and valve endothelial cells and in the current version of the manuscript the information about these two cell types is not separated. It could help the reader if the sections in which both cell types are discussed would be separated into subsections. Additionally, the three differentiation (trans-differentiation?) pathways of VICs into activated or osteoblast-like cells and VECs to mesenchymal-like cells could be discussed separately and more deeply. 

Additional figures could improve the manuscript.  

Author Response

We have acted upon these recommendations and added in a section on current trials

Reviewer 3 Report

Hanna et al. they present an interesting review in global terms and that can attract the reader's attention, with a suggestive title. However, the submitted manuscript is too simplistic. It is rather a small generalist treatise by a doctoral student.
The manuscript does not include anything new with what exists in other published reviews. The schematics presented show well-accepted mechanisms and do not appear to be proprietary.
Clinical and translational medicine aspects that could bring novelty to the manuscript are not taken into consideration.
In this sense, clinical trials and regenerative medicine should be key sections in this type of review.
I encourage the authors to rewrite the review.

Author Response

We thank the reviewer for these comments.

We feel our review is novel in its concise account regarding the haemodynamic stimuli that lead to the cellular and molecular pathways that lead to CAVD. To our knowledge, there is no other paper that has collated the most relevant papers describing this. 

Our brief for this manuscript was the pathophysiology of CAVD. While we agree a section on regenerative medicine is important, this we feel would constitute another paper regarding the management of CAVD. However, this was not the title we were given. If a section on surgical treatment and tissue engineered valves is required, to encompass regenerative medicine, we can add this in but we will need to change the title of our manuscript. 

 We have added a section on future perspectives regarding imaging and haemodynamics and its role in disease diagnosis and prognosis. WIth respect to clinical trials, throughout the manuscript we refer to studies in patients to highlight the translation of signaling pathways to the clinic. We have also added  a section on ongoing clinical trials targeting some of the molecular pathways we describe, which we hope addresses the reviewers comment regarding clinical trials. 

We have re-organised the manuscript to increase readability and added in diagrams/pictures to break up the text.

We hope the reviewer will find our changes satisfactory. 

Round 2

Reviewer 3 Report

The authors have minimally improved the presented review. The authors have made very limited changes. It is not suitable for publication.

Author Response

We thank the reviewer for the comment. 

Once again we reiterate that we have written a paper based on the title that we were given ' the pathophysiology of aortic stenosis'

We have intended to focus on haemodynamic aspects of AS, and while our review does have similarites between other reviews published recently, our review has a significant focus on haemodynamics of AS and collating the evidence of experimental studies in this area.

It is impossible to make this review completely different from the two reviews referred to by the academic editor, given the title of our review and those two reviews is the same. 

This manuscript is a resubmission of an earlier submission. The following is a list of the peer review reports and author responses from that submission.